# Effects of Phytochelatin-like Gene on the Resistance and Enrichment of Cd^2+^ in Tobacco

**DOI:** 10.3390/ijms232416167

**Published:** 2022-12-18

**Authors:** Yilin Zheng, Mengyu Li, Binman Liu, Yafei Qin, Jinhua Li, Yu Pan, Xingguo Zhang

**Affiliations:** 1Key Laboratory of Horticulture Science for Southern Mountainous Regions, The Ministry of Education, College of Horticulture and Landscape Architecture, Southwest University, Chongqing 400715, China; 2State Cultivation Base of Crop Stress Biology for Southern Mountainous Land of Southwest University, Academy of Agricultural Sciences, Southwest University, Chongqing 400715, China

**Keywords:** phytochelatin-like, cadmium, enrichment ability, tobacco, subcellular localization

## Abstract

Phytochelatins (PCs) are class III metallothioneins in plants. They are low molecular-weight polypeptides rich in cysteine residues which can bind to metal ions and affect the physiological metabolism in plants. Unlike other types of metallothioneins, PCs are not the product of gene coding but are synthesized by phytochelatin synthase (*PCS*) based on glutathione (GSH). The chemical formula of phytochelatin is a mixture of (γ-Glu-Cys)_n_-Gly (n = 2–11) and is influenced by many factors during synthesis. Phytochelatin-like (*PCL*) is a gene-encoded peptide (Met-(α-Glu-Cys)_11_-Gly) designed by our laboratory whose amino acid sequence mimics that of a natural phytochelatin. This study investigated how PCL expression in transgenic plants affects resistance to Cd and Cd accumulation. Under Cd^2+^ stress, transgenic plants were proven to perform significantly better than the wild-type (WT), regarding morphological traits and antioxidant abilities, but accumulated Cd to higher levels, notably in the roots. Fluorescence microscopy showed that PCL localized in the cytoplasm and nucleus.

## 1. Introduction

In recent decades, heavy metal pollution has become a severe environmental problem [1]. Cadmium (Cd) is a heavy metal element with substantial biological toxicity and poses a severe threat to human health since metals can enter the food chain and bioaccumulate [2,3,4,5]. Therefore, Cd-contaminated soil has raised widespread concerns. Phytoremediation is an effective and environmental alternative to physical and chemical remediation methods. Generally, plants are tolerant to low concentrations of heavy metals. However, once a certain threshold is exceeded, heavy metals have adverse effects on plant growth and development, such as the inactivation of enzymes, disruption of cell membrane integrity, inhibition of photosynthesis, root rot, and the wilting and death of plants [6,7,8,9].

Plants have evolved several detoxification mechanisms to minimize the damage caused by heavy metals, including an antioxidant system, chelating agents, and transporters [10]. Phytochelatins (PCs) were first discovered in plants and subsequently found in fungi and other organisms [11]. They are cysteine-rich polymers with the general structure (γ-Glu-Cys)_n_-Gly (n = 2–11) [12]. PCs are synthesized by plant chelating peptide synthase (PCS), using glutathione (GSH) as a substrate. Their synthesis, therefore, is not a direct product of gene expression [13]. PCs can combine with heavy metal ions to form stable high molecular weight complexes that are sequestered in vacuoles, thus reducing their adverse effects on plants [13,14]. Introducing the exogenous *BnPCS1* gene (from *Boehmeria nivea*) into tobacco significantly improved Cd endurance and antioxidant capacity under Cd stress [15]. In addition, the overexpression of *MnPCS1* and *MnPCS2* in *Arabidopsis* and tobacco enhanced Zn/Cd resistance [16].

Although some studies have shown that the overexpression of the *PCS* gene can increase Cd accumulation in plants [17], other studies have demonstrated the opposite results. Overexpression of *AtPCS1* in *Arabidopsis* leads to increased arsenic resistance and cadmium hypersensitivity [18]. Moreover, the heterologous expression of *Arabidopsis AtPCS1* in tobacco was associated with high sensitivity to Cd [19]. It has been speculated that the increase in intracellular PCs might cause a decrease in GSH, which may lead to a strong imbalance between the biosynthesis of GSH and PCs, resulting in a disruption of cellular metabolism and reduced resistance to Cd [18,19].

In this study, a phytochelatin-like (*PCL*) gene, encoding a peptide with the sequence Met-(α-G1u-Cys)_11_-Gly, was synthesized using the amino acid sequence of the plant PCs (γ-Glu-Cys)_11_-Gly as a reference. The *PCL* gene and the CaMV 35S promoter were introduced into tobacco, and the Cd endurance and accumulation ability of the transgenic tobacco plants were investigated.

## 2. Results

### 2.1. Transgenic Tobacco with PCL High Expression

The expression of the *PCL* gene in transgenic tobacco lines was analyzed by quantitative real-time PCR (qRT-PCR), and lines with high expression were selected for future experiments. The results (Figure 1) showed that *PCL* expression differed in the 13 transgenic lines. The expression levels of lines L9, L11, and L12 were about five to six times higher than that of the L1 line, which displayed the lowest expression level. In this study, the three selected lines with overexpression were designated as OE9, OE11, and OE12, and their homozygous plants were obtained from the T_2_ generation (Appendix A).

### 2.2. PCL Gene Improved the Endurance of Tobacco to Cd

We measured the root length of seedlings after germination under 0 or 200 μM Cd^2+^. Seeds of wild-type (WT) and the three homozygous transgenic lines (OE9, OE11, and OE12) were sown under 0 or 200 μM Cd^2+^ for 3 d. The results showed that the effect of Cd on root growth was evident. Under control conditions, there was no apparent difference between WT and transgenic tobacco (Figure 2a). When treated with 200 μM Cd^2+^, the root and leaf growth of transgenic and WT plants was inhibited, but the inhibition of transgenic plants was much lower than that of WT (*p* < 0.01; Figure 2b,c).

To further investigate the effect of the *PCL* gene on the growth of tobacco under Cd stress, 4-week-old transgenic and WT plants were treated with 0 or 200 μM Cd^2+^ for seven days under hydroponic conditions. The results showed that the growth of transgenic and WT plants had no noticeable difference under control conditions (Figure 3a,c), whereas the growth of WT tobacco was slow after Cd^2+^ treatment (Figure 3b,c). The root length of transgenic plants under Cd treatment was significantly longer than that of WT (Figure 3b). The leaves of WT tobacco were wilted and yellow, and the old leaves withered and dropped under Cd treatment (Figure 3c). The green loss of transgenic tobacco leaves was less severe than that of the WT, and there were no wilting symptoms (Figure 3c). Under 0 μM Cd^2+^, all chlorophyll-related parameters and fresh weight were maintained at relatively high levels, which were all significantly reduced under 200 μM Cd^2+^ conditions, including chlorophyll a, chlorophyll b, and carotenoid contents (Figure 3d–g). However, these contents in transgenic lines were higher than that in WT plants (Figure 3d–g). These results demonstrated that PCL could alleviate the impacts of Cd^2+^ stress and promote endurance to Cd^2+^.

### 2.3. PCL Gene Increases Cd Accumulation in Tobacco

To investigate the effects of *PCL* expression on Cd concentration, we determined the Cd content in the leaves and roots of transgenic and WT tobacco. Under control conditions, the Cd concentration level in transgenic and WT tobacco was very low and showed no significant difference (Figure 4a,b). Under 200 μM Cd^2+^, the leaf accumulation in WT plants was 24.1–31.6% less (OE9 and OE11, *p* < 0.01; OE12, *p* < 0.001) than in transgenic lines (Figure 4a). The Cd accumulation in the roots of transgenic plants was about 1.8–2.2 times that of WT (*p* < 0.0001; Figure 4b). The results showed that transgenic plants had a superior capacity to chelate Cd, and accumulated Cd in roots to higher levels than in aboveground parts.

### 2.4. PCL Increases Antioxidant Enzyme Activities in Tobacco

Under heavy metal stress, plant cells strengthen the antioxidant system to deal with excessive reactive oxygen species and reduce cell membrane damage. Therefore, we analyzed the catalase (CAT), peroxidase (POD), superoxide dismutase (SOD), and ascorbate peroxidase (APX) activities of transgenic and WT tobacco under Cd stress.

The results showed no significant difference in antioxidant enzyme activity between transgenic and WT plants grown under control conditions. Under 200 μM Cd^2+^ stress, the CAT activity in the leaves and roots of transgenic tobacco was 31.1–39.2% (OE9 and OE12, *p* < 0.01; OE11, *p* < 0.001) and 27.1–31.9% (OE11 and OE12, *p* < 0.05; OE9, *p* < 0.01) higher than corresponding values in WT plants (Figure 5a,b). The SOD activity in leaves and roots of transgenic tobacco was 20.7–24.4% (OE9, *p* < 0.05; OE11 and OE12, *p* < 0.01) and 28.6–35.7% higher than in WT plants (Figure 5c,d). POD activity in the leaves and roots of transgenic tobacco was 53.3–55.2% (*p* < 0.05) and 27.5–31.7% (OE9, *p* < 0.01; OE11 and OE12, *p* < 0.05) higher than in WT plants (Figure 5e,f). The APX activity in the leaves and roots of transgenic lines was about 1.5 (*p* < 0.05) and 1.2 (OE11 and OE12, *p* < 0.05; OE9, *p* < 0.01) times that of WT (Figure 5g,h). These results showed that the PCL gene enhanced the antioxidant capacity of tobacco in response to Cd stress.

### 2.5. PCL Transgenic Tobacco Was More Tolerant to Osmotic and Oxidative Stress

Under heavy metal stress, ROS levels increase, damaging various components of cells, including cell membranes, nucleic acids, and proteins [20]. Hydrogen peroxide (H_2_O_2_) is one of these ROS. H_2_O_2_ can react with 3,3′-diaminobenzidine (DAB), producing brown precipitates, and with nitrotetrazolium blue chloride (NBT), producing blue precipitates. To study the oxidative stress level in tobacco leaves under Cd stress, we measured the content of H_2_O_2_ and performed histochemical staining by DAB and NBT.

According to the results, transgenic and WT tobacco did not differ significantly in H_2_O_2_ content under control conditions. Under Cd treatment, the H_2_O_2_ content of the three transgenic lines in the leaves and roots was 28.4–32.2% (*p* < 0.0001) and 22.1–26.4% (*p* < 0.01) lower than that of the WT (Figure 6a,b). DAB and NBT histochemistry produced only weak staining in both WT and transgenic plants under control conditions. The staining intensity appreciably increased upon exposure to 200 μM Cd, with a stronger response in WT plants, reflecting higher levels of H_2_O_2_ (Figure 6c,d). The results showed that the *PCL* expression reduced the content of H_2_O_2_ in tobacco leaves under Cd stress.

Oxidative damage under heavy metal stress includes membrane lipid peroxidation. Malondialdehyde (MDA) is the final decomposition product of membrane lipid peroxidation [21]. The results showed no significant difference in MDA content between transgenic tobacco and WT under control conditions. Under 200 μM Cd^2+^, the MDA content in the roots of WT was 33.5–48.9% (*p* < 0.001) higher than in transgenic lines (Figure 7a). Likewise, the MDA content in the leaves of transgenic plants was about half that of WT plants, and only OE13 reached a level comparable with WT (*p* < 0.05; Figure 7b). The results showed that under Cd treatment, the membrane lipid peroxidation of tobacco was aggravated, and the damage was lower in transgenic than WT plants.

In addition to being indispensable for protein synthesis, proline (Pro) is also involved as an osmoprotectant and for heavy metal resistance. Pro helps maintain a correct water balance, prevents membrane distortion, and acts as a hydroxyl radical scavenger [22]. The results showed that under Cd^2+^ treatment, Pro accumulation in the leaves and roots of transgenic tobacco was 2.0–2.3 (*p* < 0.001) and 1.7–1.9 (*p* < 0.01) times that of WT (Figure 7c,d).

### 2.6. Subcellular Localization of PCL

The *PCL* was fused with the enhanced green fluorescent protein (EGFP) gene and controlled by the CaMV 35S promoter (Appendix A). The transgenic tobacco overexpression of *35S::PCL-EGFP* was cultured (Appendix A), and the protein localization was observed under laser confocal microscopy. Fluorescence showed that PCL localized both in the nucleus and the cytoplasm (Figure 8).

## 3. Discussion

Many studies have confirmed that the PC-dependent pathway is a key mechanism for plants to resist heavy metals [11,18,19,23]. PCs have the general formula (γ-Glu-Cys)_n_-Gly (n = 2–11) and are synthesized from GSH (γ-GluCysGly) by PCS [14]. The whole synthesis process includes multistep enzymatic reactions and is thus influenced by many factors. Some studies indicated that the substrate (GSH) for PCs might limit the synthesis rate. The addition of a specific GSH synthase inhibitor (buthionine sulfoximine, BSO) diminished GSH and PC accumulation in the *Dianthus carthusianorum* L. [24]. Expressing *AtPCS1* in *Arabidopsis* did not comparably affect the Cd accumulation [25]; however, expressing *GSH1* and *AsPCS1* accumulated Cd to a high level [26]. It was obvious that the presence of GSH or PCs alone was not enough to confer Cd accumulation and resistance. To construct a more efficient expression system, we synthesized PCL referring to the amino acid sequence of PCs to obtain a stable polypeptide encoded directly from a gene. Then, we expressed the *PCL* gene from the CaMV 35S promoter in transgenic tobacco to clarify its biological function.

Some studies hypothesized that expressing the PCS in plants could lead to heavy metal resistance and accumulation, but these studies presented contradictory results. Although *AtPCS1* overexpression in *Escherichia coli* [17] and *Saccharomyces cerevisiae* [27] increased Cd resistance, it resulted in hypersensitivity to Cd in *Arabidopsis* [25] and tobacco [19]. The overexpression of *AtPCS1* in tobacco (*Nicotiana tabacum)* led to increased cadmium sensitivity, while tobacco transformed with *CePCS* from *Caenorhabditis elegans* was more tolerant to Cd [19]. The functional differences between enzymes and changes in cellular thiol concentrations are possibly related to the distinct sensitivity to Cd [25]. However, no plausible explanation for these Cd resistance disparities has yet been found.

In addition, those types of PCS transformants did not show substantially increased cadmium accumulation. Similar results have been reported in other studies. Tobacco expressing *NtPCS1* from *Nelumbo nucifera* exhibited increased resistance to arsenic (As) and Cd, but changes in the accumulation of As and Cd were not significant [28]. Transgenic tobacco expressing *CdPCS1* showed Cd accumulation increased several folds after Cd exposure, but the growth was significantly inhibited [29]. The Cd content of *BnPCS* transgenic lines was significantly increased in shoots but not in roots, thus the total Cd accumulation remained low [15]. In this study, the heterologous expression *PCL* in tobacco increased the Cd accumulation to a high level. The Cd content of transgenic tobacco was 1.3–1.5 times in roots and 1.8–2.2 times in leaves that of WT, and the root accumulation was much higher than that of the aboveground part (Figure 4). Rapid Cd chelation by PCL at the root might alleviate the impacts of Cd toxicity on plants. Transgenic tobacco plants exhibited increased root length, fresh weights, chlorophyll content, Pro content, and antioxidant enzyme activities but decreased H_2_O_2_ and MDA levels. These results indicated that the *PCL* gene in tobacco could improve accumulation and resistance to Cd.

It appears that no PCS gene would be suitable for the transformation of all plant species for phytoremediation. Further research can therefore be conducted with respect to this problem. However, we have demonstrated that the *PCL* gene in tobacco could promote resistance and accumulation to Cd. In the future, we can investigate if *PCL* works on other species.

Previous studies showed that AtPCS1 [30] and VsPCS1 [31] localized in the cytoplasm. In addition to cytoplasm localization, PCS showed other kinds of cellular localization; SpPCS1 from *Schizosaccharomyces pombe* was localized to the mitochondria; and AtPCS2 [32] and BnPCS1 [15] localized in the cytoplasm and nucleus. These studies used protoplasts as material rather than intact tissue. Moreover, PCS are enzymes that play a catalytic role in the synthesis of PCs, therefore, these results only represent the possible sites for PC synthesis. In this study, the subcellular localization analysis of PCL overexpressed in transgenic tobacco showed that the PCL localized in the cytoplasm and nucleus (Figure 8). This subcellular localization indicates that PCL might be involved in other physiological processes and may merit further investigation.

## 4. Materials and Methods

### 4.1. Experimental Materials and Reagents

Wild species of bare-stem tobacco (*Nicotiana nudicaulis* Watson, 2n = 2x = 24) were provided by our laboratory. PCR Primers were synthesized by Shenzhen Huada Biotechnology Co., Ltd. (Shenzhen, China). Taq DNA polymerase and DNA Mark were from the TaKaRa company. RNA extraction reagent was purchased from Dalian Bao Biological Company. The Real-Time PCR Kit was purchased from Bio-Rad Biological Company. The reverse transcription kit, gel recovery kit, and plasmid extraction kit were purchased from Tiangen and Omega, respectively. *Escherichia coli* str. Top10, *Agrobacterium tumefaciens* str. LBA4404, Kanamycin (Kan), Streptomycin (Str), Rifampicin (Rif), Zeatin (ZT), and other antibiotics and hormones were purchased from Beijing Dingguo Changsheng Biotechnology Co., Ltd. (Beijing, China).

### 4.2. Generation of Tobacco Plants Expressing PCL

The leaves of tobacco plants grown in a sterile agar medium were used for leaf disc transformation. A sequence containing the PCL gene and *CaMV 35S* promoter (*35S-PCL*; Appendix A) was transformed into *Agrobacterium tumefaciens* strain LBA4404 by the freeze–thaw method. Tobacco leaf discs were transformed with A. *tumefaciens* [33,34].

The total RNA from tobacco leaves, isolated with RNAiso Plus (TaKaRa, Dalian, China) as described in the manufacturer’s instructions, was treated with DNase I (TaKaRa, Dalian, China). DNase-treated RNA was reverse-transcribed using a PrimeScript^TM^ RT Reagent Kit (TaKaRa, Dalian, China). qRT-PCR was performed using a CFX96 Real-Time PCR System (Bio-Rad Laboratories, Hercules, CA, USA) with Eva Green S (Bio-Rad Laboratories, Hercules, CA, USA). The relative expression of the detected gene was calculated using the 2^−∆∆Ct^ method.

Three highly expressed lines were screened by qRT-PCR using the primer pair PCL-forward (5′-CAAAGAACCAAACCCCATCGAAATAGGAG-3′) and PCL-reverse (5′-TCAGAGTCGGCCAGAAAACTCAGTTC-3′), and the Nb 18s rRNA gene was used as an internal control for normalization using the primer pair Nb 18s rRNA-forward (5′-AGTCTTTCGCTTTCTCACCATCTGCT-3′) and Nb 18s rRNA-reverse (5′-CTGCAAGAATCTCAAACACG-3′). Homozygous lines were obtained by PCR (polymerase chain reaction) from the T_2_ generation of the three highly expressed lines (Appendix A).

### 4.3. Growth Conditions and Treatments

WT and transgenic homozygous tobacco seeds were sterilized with 20% sodium hypochlorite for 10 min and then germinated in an incubator at 28 °C. After germination, the seeds were arranged in Petri dishes (each containing 30 seeds and three replicates) with a layer of filter paper and 20 mL of Hoagland nutrient solution [35] containing 0 or 200 μM CdCl_2_. The Petri dish was placed vertically for 3 days to allow the roots of the tobacco seedlings to grow vertically. Root growth and differences between tobacco lines were observed and statistically recorded.

The treatment method for adult plants is not the same as for seedlings. The WT and the homozygous lines of the transgenic tobacco were sown in vermiculite. After germination, the seedlings were transferred to Hoagland solution at 26 °C and the light cycle was adjusted to 16 h daylight (250 μmol m^−2^ s^−1^) and 8 h night. When the plants were 4 weeks old, transgenic tobacco plants and WT plants with the same growth status were transferred to Hoagland nutrient solution containing either 0 or 200 μM CdCl_2_. After 7 days, leaf and root samples were collected for analysis of physiological indexes, with at least three samples repeated in each group.

### 4.4. Determination of Cadmium in Plants

After being treated with 0 and 200 μM Cd^2+^ for seven days, the samples were washed with distilled water for a short time and then dried separately at 60 °C. The content of Cd in dried specimens was determined by atomic absorption spectrometry [36].

### 4.5. Determination of Chlorophyll Content in Plants

In total, 0.1 g of fresh samples were cut and placed in 5 mL of 95% ethanol and soaked in the dark for 24 h. The absorbance of the pigment extract was measured at 665 nm, 649 nm, and 470 nm [37].

### 4.6. Measurement of Antioxidant Enzyme Activity

Fresh leaf and root samples (0.2 g) were harvested and thoroughly ground in 1.6 mL 0.05 mol L^−1^ (pH = 7.8) precooled phosphate buffer with some quartz sand grains. The homogenate was centrifuged at 12,000 rpm at 4  °C, centrifuged for another 15 min, and then the supernatant was retained.

The total SOD was estimated from the increase in NADH oxidation and measured at 560 nm using its molar extinction coefficient of 6.22 × 10^3^ M^−1^ cm^−1^ [38].

POD activity was measured by monitoring the guaiacol dehydrogenation product at 436 nm (extinction coefficient 6.39 mM^−1^ cm^−1^) [39].

The activity of CAT was determined by measuring the decomposition of H_2_O_2_ at 240 nm (molar extinction coefficient 39.4 M^−1^ cm^−1^) [40].

APX activity was estimated by measuring the absorbance at 290 nm ascorbate (molar extinction coefficient 2.88 mM^−1^ cm^−1^) [40].

### 4.7. Determination of MDA and Pro Content in Plants

Pro and MDA were measured by acid-ninhydrin and colorimetric methods, respectively [41]. The MDA content was determined in 0.1 g of the samples using 4 mL of thiobarbituric acid (TCA)/thiobarbiturate (TBA) reagent {0.67% (*w*/*v*) TBA in 10% TCA (*w*/*v*)}. Absorbance was measured at 450, 535, and 600 nm, and the MDA concentration was calculated using an extinction coefficient of 155 mM^−1^ cm^−1^. Proline was extracted with 3% sulfosalicylic acid from 0.1 g of plant tissue and determined with ninhydrin reagent. The absorbance was measured at 520 nm (extinction coefficient 8.7 × 106 M^−1^ cm^−1^).

### 4.8. Determination of H_2_O_2_ Content

H_2_O_2_ content was determined by homogenizing the samples (0.5 g) with 2.5 mL of 0.1% (*w*/*v*) TCA and centrifuging at 13,000× *g* for 30 min. The reaction mixture consisted of 0.5 mL supernatant, 0.5 mL (0.1 M) potassium phosphate buffer (pH 7.6), and 2 mL (1 M) KI. After 1 h of incubation in the dark, the absorbance was measured at 390 nm, and the H_2_O_2_ content was calculated using a standard curve for H_2_O_2_ [41].

### 4.9. DAB and NBT Staining

Leaf samples were collected after seven days of Cd treatment, washed with distilled water, and immersed in 1% DAB (50 mM Tris-HCl, pH 3.8) and 0.5 mg mL^−1^ NBT (25 mM HEPES, pH 7.8) for 6 h in the dark. The leaves were then transferred to anhydrous ethanol and bathed in boiling water for about 10 min until the chlorophyll disappeared utterly [42].

### 4.10. Subcellular Localization of PCL Peptide

The *PCL* gene was fused with the EGFP gene and controlled by the CaMV 35S promoter in a binary vector (Appendix A). Transgenic tobacco plants with the integrated *35S::PCL-EGFP* gene were obtained (Appendix A). Localization of PCL-EGFP was observed by laser confocal microscopy.

### 4.11. Statistical Analysis

All the tests were repeated three times independently. The measured data were statistically analyzed using Microsoft Excel (version 2019) and graphically displayed GraphPad Prism (GraphPad Software 8.0.2, San Diego, CA, USA).

## 5. Conclusions

We synthesized PCL according to the amino acid sequence of PCS and examined its function in the Cd resistance of tobacco. We found that the PCL localized in both the nucleus and cytoplasm and that the heterologous expression of the artificial PCL significantly improved plant Cd resistance and accumulation.

## Figures and Tables

**Figure 1 ijms-23-16167-f001:**
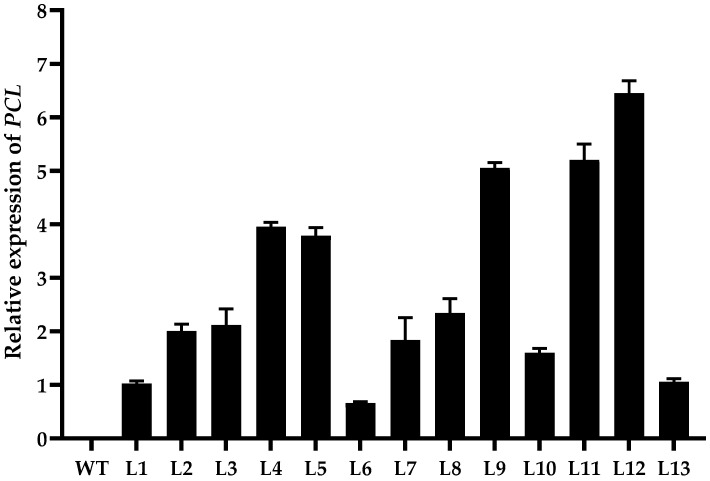
*PCL* transcript levels between different lines. WT: wild-type. L1–L13: *PCL* transgenic lines of *Nicotiana nudicaulis*.

**Figure 2 ijms-23-16167-f002:**
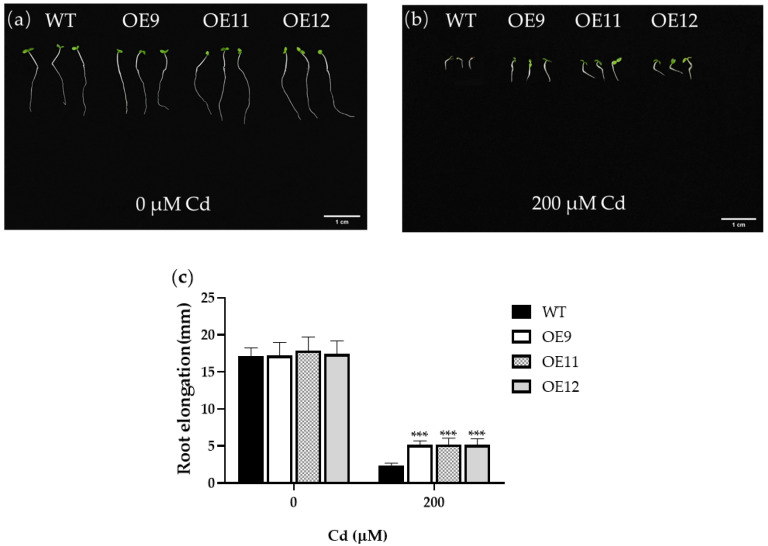
Effects of Cd stress in transgenic and WT seedlings. (**a**) Phenotypic comparison of the WT and transgenic lines under 0 μM Cd^2+^. (**b**) Phenotypic comparison of WT and transgenic lines under 200 μM Cd^2+^. (**c**) Root elongation of the WT and transgenic seeds (*n* = 30). WT: Wild-type tobacco. OE9, OE11, and OE12: Three homozygous transgenic tobacco lines (T_2_). Transgenic and WT seedlings were grown in Hoagland nutrient solution containing 0 or 200 μM Cd^2+^ for 3 d. Asterisks indicate the significant difference between OE lines and WT (***, *p* < 0.001).

**Figure 3 ijms-23-16167-f003:**
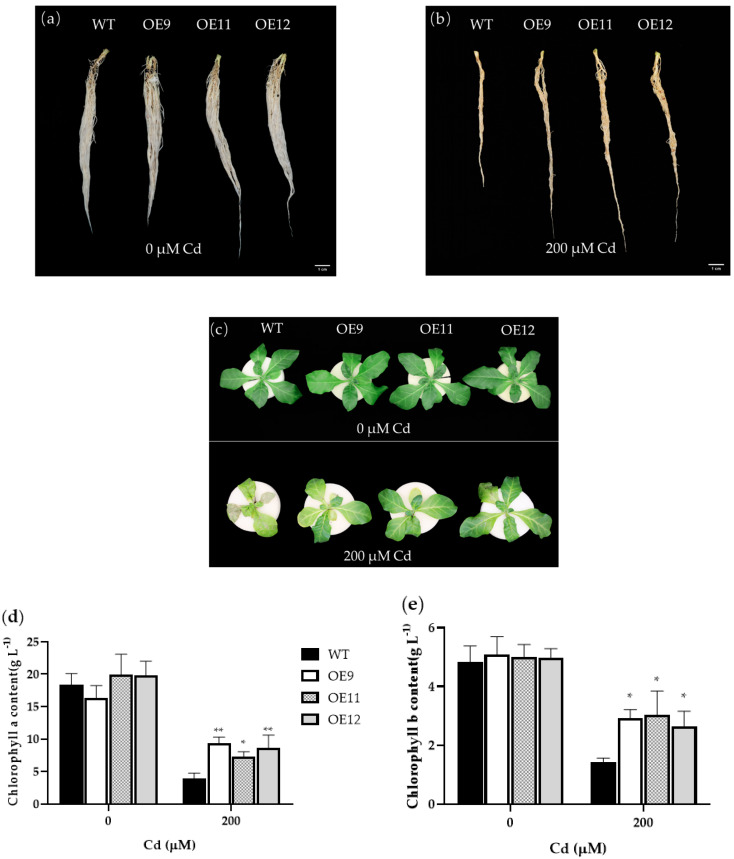
Effects of Cd stress in transgenic and WT plants. (**a**) Root phenotype of the WT and transgenic lines under 0 μM Cd^2+^. (**b**) Root phenotype of the WT and transgenic lines under 200 μM Cd^2+^. (**c**) Leaf phenotype in the WT and transgenic lines under 0 or 200 μM Cd^2+^. (**d**) Chlorophyll a content in WT and transgenic plants. (**e**) Chlorophyll b content in WT and transgenic plants. (**f**) Carotenoid content of the WT and transgenic plants. (**g**) Fresh weight of the WT and transgenic plants. Data represent the means ± SD (*n* = 3). WT: Wild-type tobacco. OE9, OE11, and OE12: Homozygous transgenic tobacco lines (T_2_). The 4-week-old transgenic lines and WT plants were grown in Hoagland nutrient solution containing 0 or 200 μM Cd for 7 d. Asterisks indicate a significant difference between the transgenic lines and the WT (*, *p* < 0.05; **, *p* < 0.01; ***, *p* < 0.001).

**Figure 4 ijms-23-16167-f004:**
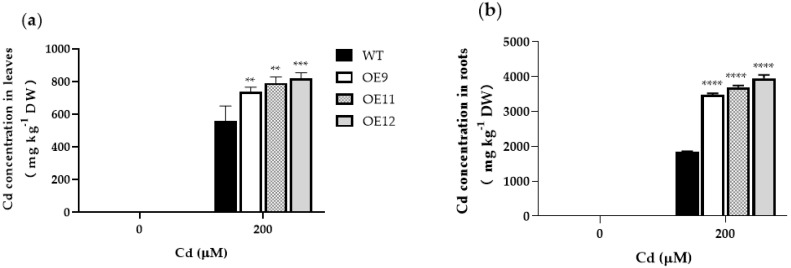
Cd accumulation in wild-type (WT) and transgenic tobacco plants (OE9, OE11, and OE12). (**a**) Cadmium concentrations in leaves. (**b**) Cadmium concentrations in the roots. The 4-week-old transgenic lines and WT plants were grown in Hoagland nutrient solution containing 0 or 200 μM Cd for 7 d. Data represent the means ± SD (n = 3). Asterisks indicate a significant difference between the transgenic lines and the WT (**, *p* < 0.01; ***, *p* < 0.001; ****, *p* < 0.0001).

**Figure 5 ijms-23-16167-f005:**
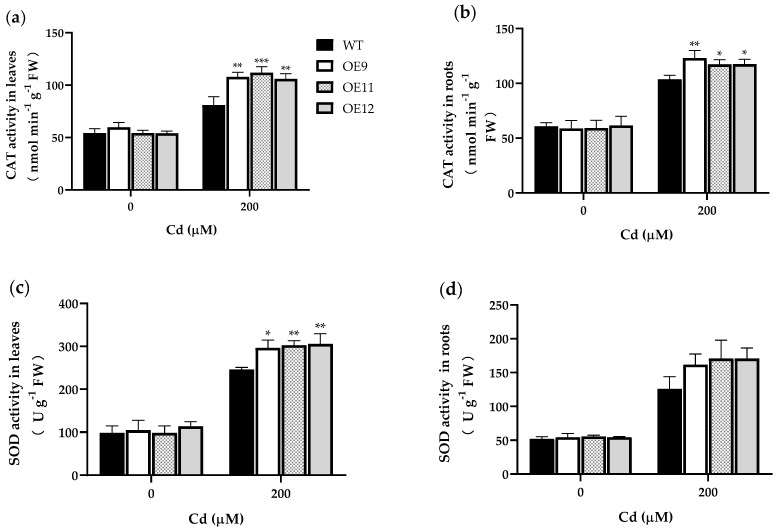
Antioxidant enzyme activities in the leaves and roots of wild-type (WT) and transgenic tobacco plants (OE9, OE11, and OE12). (**a**) CAT activities in leaves. (**b**) CAT activities in roots. (**c**) SOD activities in leaves. (**d**) SOD activities in roots. (**e**) SOD activities in leaves. (**f**) SOD activities in roots. (**g**) APX activities in leaves. (**h**) APX activities in roots. Data represent the means ± SD (*n* = 3). The 4-week-old transgenic lines and WT plants were grown in Hoagland nutrient solution containing 0 or 200 μM Cd for 7 d. Asterisks indicate a significant difference between the transgenic lines and the WT (*, *p* < 0.05; **, *p* < 0.01; ***, *p* < 0.001).

**Figure 6 ijms-23-16167-f006:**
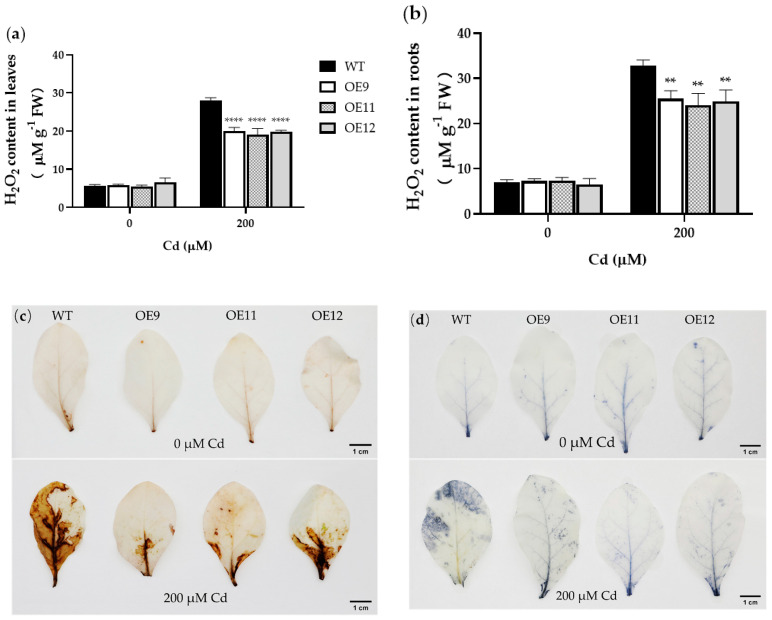
H_2_O_2_ content, DAB, and NBT staining under 0 or 200 μM Cd in wild-type (WT) and transgenic tobacco plants (OE9, OE11, and OE12). (**a**) H_2_O_2_ content in leaves. (**b**) H_2_O_2_ content in roots. (**c**) DAB staining. (**d**) NBT staining. Data represent the means ± SD (*n* = 3). Asterisks indicate a significant difference between the transgenic lines and the WT (**, *p* < 0.01; ****, *p* < 0.0001).

**Figure 7 ijms-23-16167-f007:**
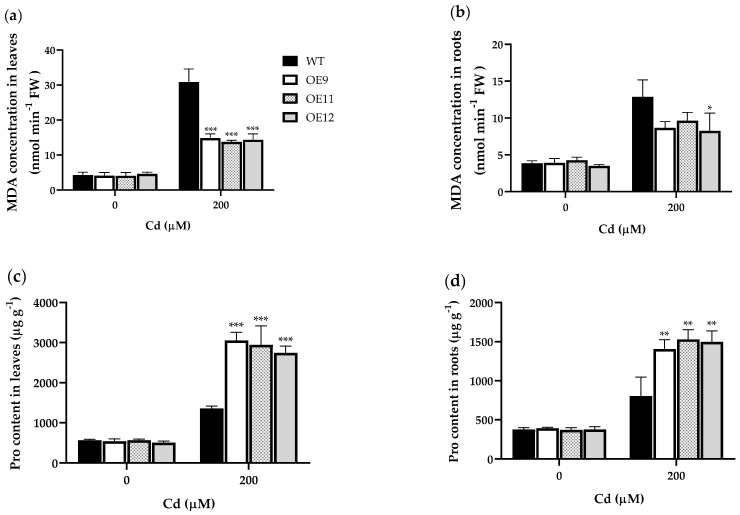
MDA and Pro content under 0 or 200 μM Cd in wild-type (WT) and transgenic tobacco plants (OE9, OE11, and OE12). (**a**) MDA content in leaves (**b**) MDA content in roots. (**c**) Pro content in leaves (**d**) Pro content in roots. Data represent the means ± SD (*n* = 3). Asterisks indicate a significant difference between the transgenic lines and the WT (*, *p* < 0.05; **, *p* < 0.01; ***, *p* < 0.001).

**Figure 8 ijms-23-16167-f008:**
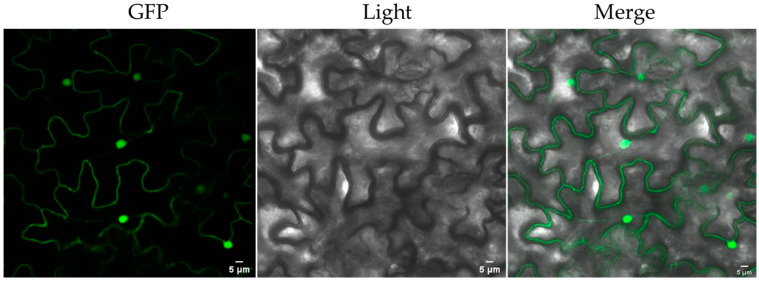
Subcellular localization of the PCL protein in the leaf epidermal leaf cells of a transgenic tobacco plant. Images obtained by Leica confocal microscopy. GFP: GFP fluorescence under green light; Light: visible light image; Merge: merged images.

## Data Availability

Not applicable.

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
