# Peer review of "Effects of Phytochelatin-like Gene on the Resistance and Enrichment of Cd2+ in Tobacco"

_ijms, 2022, doi:10.3390/ijms232416167_

Round 1

Reviewer 1 Report

Effects of Phytochelatin-like Gene on the Endurance and Enrichment of Cd2+ in Tobacco

Yilin Zheng 1,2 , Mengyu Li 1,2 , Binman Liu 1,2 , Yafei Qin 1,2 , Jinhua Li 1,2 , Yu Pan 1,2 , Xingguo Zhang

The paper investigates how the expression of an artificial gene encoding for a phytochelatin-like peptide affects the response to Cd stress in tobacco plants. The study provides robust evidence that (1) transgenic plants are more resistant than wild plants to adverse effects of Cd, and (2) transgenic plants accumulate larger amounts of Cd in their tissues, particularly in the roots.

The paper is of interest and deserves publication. I have provided an English revision in the attached pdf, but the Discussion section must be completely rewritten. In the present form it merely reiterates the results. The authors should provide a succinct summary of the results and discuss them in the context of current insight. In my opinion, figure 8 shows that PCL localizes in cell walls (not the cytoplasm) besides the nucleus. Both localizations need being addressed. A further point worth of attention is that PCL gene expression improves plant resistance to Cd, but also enhances Cd accumulation in plant tissues, the latter effect raising doubts on the potential application of the technique.

Further points:

The title uses the neutral term “endurance”, whereas the text uses “tolerance”. I feel that “resistance” is more correct when referring to transgenic plants. I suggest modifying the title as follows:

Expression of a Phytochelatin-like Gene Enhances Resistance to Cd Stress and Increases Cd Accumulation in Tobacco.

Lines 24-26: the sentence is obscure. Cd is an element that can only exist as Cd+2 under natural conditions, and there is no way of metabolizing it.

Line 72 Generation of tobacco plants expressing PCL: give more details about the procedure adopted, notably the starting plant material (protoplasts?)

Lines 97:  the authors state that the treatment lasted 7 days, yet the data reported in Figs 2 refer to a 72 h treatment (lines 190).

Line 110: rewrite

Line 183: Fig 2b does not show the leaf color: delete reference

Lines 240-244: irrelevant and repetitive, delete

Line 300: give full name of MDA

Author Response

Point 1: The paper investigates how the expression of an artificial gene encoding for a phytochelatin-like peptide affects the response to Cd stress in tobacco plants. The study provides robust evidence that (1) transgenic plants are more resistant than wild plants to adverse effects of Cd, and (2) transgenic plants accumulate larger amounts of Cd in their tissues, particularly in the roots.

The paper is of interest and deserves publication. I have provided an English revision in the attached pdf, but the Discussion section must be completely rewritten. In the present form it merely reiterates the results. The authors should provide a succinct summary of the results and discuss them in the context of current insight.

Response 1: We find these comments and the English revision very insightful and useful. After considering these important suggestions and revising the manuscript accordingly, we hope the revisions are satisfactory.

Point 2: In my opinion, figure 8 shows that PCL localizes in cell walls (not the cytoplasm) besides the nucleus. Both localizations need being addressed. A further point worth of attention is that PCL gene expression improves plant resistance to Cd, but also enhances Cd accumulation in plant tissues, the latter effect raising doubts on the potential application of the technique.

Response 2: Thanks for the reviewer’s helpful comments. After reading other studies, we still believe that PCL localizes in the cytoplasm. We found a similar subcellular localization in a paper (DOI: 10.1111/nph.16860). And we put the figure in attached word (respond to reviewer1), hope this will be able to answer this question.

As for the doubts that PCL enhanced Cd accumulation in plants, we believe this is precisely the advantage of PCL as a potential gene for phytoremediation to remediate soil heavy metal. We are also trying to find mechanisms (transporters or particular promoters) to trap heavy metals in the roots without transporting them to the aboveground parts. If we implement this idea, it will allow for a broader application.

Point 3: The title uses the neutral term “endurance”, whereas the text uses “tolerance”. I feel that “resistance” is more correct when referring to transgenic plants. I suggest modifying the title as follows: Expression of a Phytochelatin-like Gene Enhances Resistance to Cd Stress and Increases Cd Accumulation in Tobacco.

Response 3: Thank you for the suggestion. We agree that “resistance” is more suitable in this study.

We have modified the title and text.

Point 4: Lines 24-26: the sentence is obscure. Cd is an element that can only exist as Cd+2 under natural conditions, and there is no way of metabolizing it.

Response 4: Thank you for pointing this out. We have rewritten the section.

Point 5: Line 72 Generation of tobacco plants expressing PCL: give more details about the procedure adopted, notably the starting plant material (protoplasts?)

Response 5: Thank you for the comment. We used the leaf disc transformation method, and we have added the details of the transformation in Sub-section 4.2.

Point 6: Lines 97: the authors state that the treatment lasted 7 days, yet the data reported in Figs 2 refer to a 72 h treatment (lines 190).

Response 6: We first observed the Cd resistance at the seedling stage and measured the root length. Then We used the adult plants for further research. Since we found the degree of Cd resistance was different between seedlings and adult plants, 3 days of Cd treatment for seedlings and 7 days for adult plants. We mentioned the seedlings treatment in the Sub-section “Measurement of root length” in the previous manuscript. We have rewritten it in Sub-section 4.3 for a clearer presentation.

Point 7: Line 110: rewrite

Response 7: Thank you for the comment. We have rewritten the sentence.

Point 8: Line 183: Fig 2b does not show the leaf color: delete reference

Response 8: Thank you for pointing this out. We have deleted the reference

Point 9: Lines 240-244: irrelevant and repetitive, delete

Response 9: Thank you for pointing this out. We have deleted Lines 240-244.

Point 10: Line 300: give full name of MDA

Response 10: Thank you for the comment. We have revised the full name of MDA.

Reviewer 2 Report

This study intends to investigate the role of PCL in Cd tolerance of tobacco. They synthesized PCL and overexpressed it in tobacco. However, the author should introduce the method they used in the PCL synthesizing or cloning in the Materials and Methods section. And the alignment between PCS1 and PCL need to be revealed to show the unique function of PCL to PCS1.  In addition, the tobacco with PCL knock-out or inhibition should be generated to completely reveal the function of PCL.

The writing and grammar are poor, should be reviewed and polished. such as,

The Abstract is not well described, the background is too much, authors should show the important results of the study in the abstract.

Line 29  the sentence is incomplete,

Line 32 should be products,

Line33 inactive enzyme

Line 49 line 52 …. Arabidopsis in italic or arabidopsis

Line74 FigureS1b or FigureS1a?  the order of figures cited in the text.  the expression of the wild-type should be added in the Figure 1

the homozygous plants can not be obtained in the T2 generation, in the T3

Figure 2c y-axis Root elongation

Figure 3 the growth indices, such as fresh weight, dry weight or water content of root and leaf should also be detected to describe the phenotype under Cd stress

Figure 4 the concentration of Cd under normal conditions was not shown

Author Response

Point 1: This study intends to investigate the role of PCL in Cd tolerance of tobacco. They synthesized PCL and overexpressed it in tobacco. However, the author should introduce the method they used in the PCL synthesizing or cloning in the Materials and Methods section. And the alignment between PCS1 and PCL need to be revealed to show the unique function of PCL to PCS1. In addition, the tobacco with PCL knock-out or inhibition should be generated to completely reveal the function of PCL.

Response 1: Thanks for the comment. In this study, PCL was synthesized referring to the amino acid sequence of PCs, and PCs ((γ-Glu-Cys)n-Gly) are synthesized from GSH (γ-GluCysGly) by PCS. As catalytic enzymes, the sequences and structures of PCS are not relevant to PCL. We agree that knock-out and inhibition generation could completely reveal the gene function. PCL is not naturally present in tobacco but is an artificially synthesized gene, so we can’t get knock-out or inhibition plants.

Point 2: The writing and grammar are poor, should be reviewed and polished. such as,

The Abstract is not well described, the background is too much, authors should show the important results of the study in the abstract.

Response 2: Thanks for the reviewer’s helpful comments. After considering these important comments, we have rewritten the manuscript. We hope the revisions are satisfactory.

Point 3: Line 29  the sentence is incomplete,

Response 3: Thanks for your comment. We have revised the sentence.

Point 4: Line 32 should be products,

Response 4: Thanks for the comment . We have revised the word.

Point 5: Line33 inactive enzyme

Response 5: Thanks for the comment . We have revised the sentence.

Point 6: Line 49 line 52 …. Arabidopsis in italic or arabidopsis

Response 6: Thanks for pointing out this important point. We have revised the word.

Point 7: Line74 FigureS1b or FigureS1a?  the order of figures cited in the text.  the expression of the wild-type should be added in the Figure 1

Response 7: Thank you for the comment. We have rewritten Sub-section 4.2 and put the right order of figures cited in the text. We have added the expression of the wild-type in the Figure 1.

Point 8: the homozygous plants can not be obtained in the T2 generation, in the T3

Response 8: Thank you for pointing this out. We called the first generation T0, I think maybe our T2 generation means the T3. We have changed T2 to T3 if necessary.

Point 9: Figure 2c y-axis Root elongation

Response 9: Thanks for the comment. We have revised the Y-axis.

Point 10: Figure 3 the growth indices, such as fresh weight, dry weight or water content of root and leaf should also be detected to describe the phenotype under Cd stress

Response 10: We thank the reviewer for pointing out the deficit. To determine the Cd accumulation, we measured the fresh and dry weight of leaves and roots, respectively. To concisely present the results, we added the total fresh weight to Figure 3.

Point 11: Figure 4, the concentration of Cd under normal conditions was not shown

Response 11: Thanks for the comment. Since the normal condition was control treatment without Cd stress, Cd content was barely detectable.

Round 2

Reviewer 2 Report

Please check the text editing carefully and then the manuscript can be acceptted. Thank you